# Literature Review: The sFlt1/PlGF Ratio and Pregestational Maternal Comorbidities: New Risk Factors to Predict Pre-Eclampsia

**DOI:** 10.3390/ijms24076744

**Published:** 2023-04-04

**Authors:** Nataliia Sergeevna Karpova, Olga Pavlovna Dmitrenko, Tatyana Sergeevna Budykina

**Affiliations:** 1Federal State Budgetary Institution “Research Institute of Pathology and Pathophysiology”, St. Baltiyskaya, House 8, Moscow 125315, Russia; 2State Budgetary Health Institution of the Moscow Region “Moscow Regional Research Institute of Obstetrics and Gynecology”, St. Pokrovka, d.22a, Moscow 101000, Russia

**Keywords:** pre-eclampsia, sFlt1/PlGF ratio, maternal comorbidities, soluble Fms-like tyrosine kinase-1, COVID-19, cardiovascular disease, pregnancy, chronic hypertension, HELLP

## Abstract

One of the main causes of maternal and neonatal morbidity and mortality is pre-eclampsia. It is characterized by a high sFlt1/PlGF ratio, according to prior research. Pregestational diseases in mothers may increase the risk of developing pre-eclampsia. Only a few studies have looked at the connection between maternal comorbidities before conception and the sFlt1/PlGF ratio. The most recent information regarding the association between maternal pregestational diseases and the ratio of sFlt1/PlGF is described in this review. The paper also examines current research suggesting that changes in pregnancy hormones and metabolites are related to a high sFlt1/PlGF ratio. Certain maternal disorders have been found to dramatically raise sFlt-1 and sFlt1/PlGF levels, according to an analysis of the literature. There is still debate about the data on the association between the sFlt1/PlGF ratio and maternal disorders such as HIV, acute coronary syndromes, cardiovascular function in the mother between 19 and 23 weeks of pregnancy, thyroid hormones, diabetes, and cancer. Additional research is needed to confirm these findings.

## 1. Pre-Eclampsia

Pre-eclampsia is a disorder of pregnancy associated with new-onset hypertension, which occurs most often after 20 weeks of gestation and frequently near term. Although often accompanied by new-onset proteinuria, hypertension and other signs or symptoms of pre-eclampsia may present in some women in the absence of proteinuria [1]. Pre-eclampsia develops in 3–8% of pregnant women and is among the top five causes of maternal morbidity and mortality, which can occur in 9–26% of cases [2,3]. If women with this disease are swiftly diagnosed, screened, and successfully treated, the majority of pre-eclampsia-related deaths can be prevented [3].

The pathogenetic mechanisms and etiology of pre-eclampsia are not yet fully understood. Pre-eclampsia has been associated with impaired placental dysfunction in early pregnancy, followed by generalized inflammation and progressive endothelial dysfunction. The pathogenic factor in the development of the disease is an imbalance of angiogenic and anti-angiogenic proteins during pregnancy [1,4].

Pre-eclampsia is commonly divided into early and late phases. Early pre-eclampsia (early-onset pre-eclampsia, EOPE) develops before 34 weeks of gestation and accounts for 5–20% of all cases of pre-eclampsia. This subtype of pre-eclampsia is also called “placental pre-eclampsia”.

Late pre-eclampsia (late-onset pre-eclampsia, LOPE) occurs after 34 weeks of gestation and accounts for >80% of all cases. This is the maternal syndrome (also called “maternal pre-eclampsia”) [5]. EOPE is commonly associated with early pregnancy placental dysfunction and subsequent growth restriction, while LOPE is connected to maternal endothelial dysfunction, which is associated with an imbalance of angiogenic and antiangiogenic mediators, in particular, placental growth factor and soluble fms-like tyrosine kinase-1(sFlt-1) [6,7].

## 2. sFlt-1 and PlGF

Anti-angiogenic sFlt-1 is encoded by the *FLT1* gene located on chromosome 13. There are several splice variations in total sFlt-1. Moreover, e15a is expressed by the placenta, inhibits VEGF, and is significantly elevated in patients with pre-eclampsia. The placenta appears to be the main source of elevated sFlt-1 in pre-eclampsia, as within 48 h of birth, the circulating levels are dramatically reduced [8]. 

The angiogenic placental growth factor (PlGF) is a 25-kd angiogenic protein that is produced in the human placenta and encoded by the *PGF* gene located on chromosome 14. Four splice isoforms of the single PlGF gene, PlGF-1 (PlGF131), PlGF-2 (PlGF152), PlGF-3 (PlGF203), and PlGF-4 (PlGF224), are produced in humans [9,10,11]. PlGF has a wide range of non-redundant effects in stressful circumstances, including sepsis, tissue ischemia, and wound healing [12]. PlGF is prominently raised in circulation during pregnancy. The role of PlGF in the placenta is thought to involve promoting growth, maturation of the vascular system, and proliferation of trophoblast cells [12,13].

## 3. sFlt1/PlGF Ratio

The first data confirming the hypotheses about the imbalance of angiogenic and antiangiogenic proteins in pre-eclampsia patients were collected when Levine et al. (2004) found that, in pre-eclampsia, the levels of sFlt-1 increased while those of PlGF decreased [14]. In the following few years, these data have been repeatedly confirmed [7,15,16,17,18,19,20,21]. 

Dramatically increasing the sFlt1/PlGF ratio can lead to pathological changes that are typical to pre-eclampsia: placental abruption [22], placental hypoxia [23], ischemic placental disease [24], placental vascular lesions [25], oxidative stress [26,27], endoplasmic reticulum stress [23,28], and maternal endothelial sensitization [23] (Figure 1). These changes can lead to ectopic pregnancy and a missed abortion [29] but do not necessarily lead to them. Tikkanen et al. did not confirm the data that a high sFlt1/PlGF ratio can predict placental abruption, and this issue is still being discussed [22,30].

Some studies, such as the ones conducted by Alahakoon et al. and Herraiz et al., have shown that pre-eclampsia, fetal or intrauterine growth restriction (FGR/IUGR), and HELLP (hemolysis, elevated liver enzymes and low platelets) have similar angiogenic profiles [16,31]. An assessment of the sFlt1/PlGF ratio makes it possible to distinguish pregnant women with probable pre-eclampsia and IUGR from women with signs of ongoing complications from other hypertensive disorders of pregnancy [20,32,33] and to confirm their diagnosis [34]. In terms of the severity of hypertensive disorders of pregnancy and IUGR, the levels of circulating angiogenic agents are the highest shortly before delivery and correlate with the severity of pre-eclampsia [17,34,35,36].

## 4. Short-Term Pre-Eclampsia Predictions Using the sFlt1/PlGF Ratio

Using the sFlt1/PlGF ratio may help to make short-term predictions about the development of the disease pathology and also serves as a reason for immediate hospitalization (Figure 2) [37,38].

According to studies by Zaisler et al., an sFlt-1/ PlGF ratio < 38 allows excluding the development of pre-eclampsia in the following 4 weeks (NPV 99.3% in the next 1 week, NPV 97.9% in the next 2 weeks and NPV 94.3% in the next 4 weeks) [39,40]. Pre-eclampsia in the following week would not require pregnancy termination if the sFlt1/PlGF ratio was 38 or lower [41] (Figure 2A,D). 

An sFlt1/PlGF ratio of 38–85 for EOPE or 38–110 for LOPE indicates a high risk of developing pre-eclampsia and/or a negative pre-eclampsia-related outcome in the following 4 weeks [40]. The cut-off value of the sFlt1/PlGF ratio at 85 is the point at which EOPE could be diagnosed with a sensitivity of 89% and a specificity of 94%. In patients with other hypertension disorders, the cut-off value of 85 is typically not exceeded during pregnancy [42]. It was later determined that the cut-off sFlt1/PlGF ratio for LOPE should be greater than 110 [43]. Repeated measurements of the sFlt1/PlGF ratio a week later may be useful to track changes in angiogenic factors and the risk of pre-eclampsia [44,45]. In individuals with sFlt1/PlGF ratios of more than 85, any unfavorable maternal outcomes, such as HELLP syndrome, increased liver enzymes, thrombocytopenia, placental abruption, or acute kidney injury, could be present (Figure 2B,E) [46]. 

Women who have an sFlt1/PlGF ratio greater than 85 for EOPE and greater than 110 for LOPE should be followed up individually or hospitalized [44]. The probability of a late abortion in asymptomatic pregnant women is independently correlated with sFlt-1 and PlGF (Figure 2C,F) [47]. Despite this, Costa et al. concluded that the sFlt1/PlGF ratio should only be used to exclude pre-eclampsia at 20–36 weeks of gestation and cannot replace the standard approaches to the diagnosis of the disease [48]. 

Taking these data together confirmed that using the ratio is a powerful prognostic and diagnostic tool for both EOPE and LOPE in routine clinical practice. 

## 5. sFlt1/PlGF Ratio and HELLP

Pre-eclampsia and HELLP syndrome may occur simultaneously, increasing the chance of negative outcomes for the mother and fetus [49]. The 2014 ISSHP statement indicates that HELLP syndrome should be suspected when the platelet count is low (less than 100 G/L), the aspartate aminotransferase (AST) or alanine aminotransferase (ALT) levels are higher than twice the upper limit of normality for the local laboratory (typically > 70 U/L), and the lactic acid dehydrogenase (LDH) level increases by more than 600 U/L as a result of hemolysis [50,51]. Patients with both early-onset and late-onset HELLP syndrome have higher levels of sFlt-1 and lower PlGF levels compared to healthy controls [52]. The highest sFlt1/PlGF ratios were seen in pre-eclampsia and HELLP patients, equaling 282 (7–948), while isolated HELLP cases had the lowest ratios and sFlt-1 values, equaling 49 (3–405). Thus, isolated HELLP syndrome appears to be a distinct disease with a distinctive angiogenic behavior from either classical pre-eclampsia or pre-eclampsia accompanied by HELLP syndrome [50]. 

In women with acute fatty liver of pregnancy (AFLP), serum levels of sFlt-1 and the sFlt1/PlGF ratio rise even higher than those in women with HELLP syndrome. In one of the patients from Suzuki’s study, serum levels of sFlt-1 (273,040 pg/mL compared to 15,135 pg/mL) and sFlt1/PlGF ratio (4236 compared to 224) were much greater than those in people with HELLP syndrome. The PlGF serum levels were comparable to those of people with HELLP syndrome [53]. 

Thus, AFLP and HELLP syndromes may exacerbate pre-eclampsia by further increasing sFlt-1 levels.

## 6. sFlt1/PlGF and Maternal Pregestational Comorbidities

A number of researchers have found that maternal pregestational comorbidities (chronic hypertension, cardiovascular disease, rheumatoid arthritis, chronic kidney disease, diabetes, COVID-19, HIV infection, thyroid hormones, cigarette smoking, and cancer) may enhance the likelihood of having a high sFlt1/PlGF ratio. The *FLT1* gene is expressed not only in the placenta but also in other tissues, such as the muscles (especially smooth muscles), female tissue, skin, male tissue, and the respiratory system [54,55]. We hypothesize that the increased expression of the *FLT1* gene in these tissues might be associated with the development of pregestational complications. This is because of the overexpression of sFlt-1, which can increase the sFlt1/PlGF ratio during pregnancy and lead to pre-eclampsia. 

## 7. sFlt1/PlGF Ratio and Chronic Hypertension

Chronic arterial hypertension (CH) is defined as the onset of an increase in blood pressure to blood pressure (BP) ≥ 140/90 mmHg. Art before pregnancy or during the first 20 weeks of pregnancy, which usually persists for more than 12 weeks after delivery [56,57]. This complication occurs in 1–2% of pregnancies. Women with CH are at an increased risk of developing pre-eclampsia [58,59]. 

There is limited and inconsistent evidence supporting the usefulness of the sFlt1/PlGF ratio [60,61]. The results of studies by Perni et al. show higher sFlt1/PlGF ratios in pregnant women with superimposed EOPE compared to EOPE without CH [61]. Costa et al. found no significant differences in the sFlt1/PlGF ratio between women with chronic hypertension and superimposed pre-eclampsia before 32 weeks gestation. However, their study had limitations, such as a small sample size and lack of normotensive control of pregnant women [62]. In women with chronic hypertension, and especially in those who subsequently develop pre-eclampsia, both PlGF and sFlt-1 concentrations are reduced in the first trimester. In addition, the PlGF and sFlt-1 levels provided a poor predictor of superimposed pre-eclampsia [58].

Further research on the potential role of angiogenic markers in the prediction and diagnosis of pre-eclampsia superimposed on chronic hypertension is needed to determine their clinical usefulness.

## 8. sFlt1/PlGF Ratio and Cardiovascular Disease

Cardiovascular diseases (CVD) are a group of diseases of the heart and blood vessels and are the leading cause of morbidity and mortality worldwide [63]. 

Alterations in maternal or fetal cardiovascular function have also been linked to pregnancy-related abnormalities in PlGF and sFlt-1. Furthermore, *FLT1* is expressed in tissues other than the placenta, such as in muscles (particularly smooth muscles, including heart and blood vessels), not only during pregnancy [54,55]. Mothers with pre-eclampsia may have elevated levels of sFlt-1, including in the heart, based on the assumption that genetic variants in the *FLT1* gene region affect its expression. This can result in an imbalance of angiogenic and anti-angiogenic proteins, raising the risk of CVD even before pregnancy. As a result, women with CVD may be at a higher risk of pre-eclampsia.

The majority of research on the relationship between CVD and angiogenic factors has been conducted on non-pregnant women. PlGF levels that are elevated during the acute phase of myocardial infarction are associated with poorer outcomes [64]. Only sFlt-1, but not PlGF, may predict poor clinical outcomes in patients with symptoms suggestive of an acute myocardial infarction [65]. Patients with heart failure have higher sFlt-1 levels, and this protein significantly predicts severe cardiac events and worse outcomes, particularly when combined with conventional biomarkers [66,67]. The PlGF/sFlt-1 ratio in patients with stable coronary artery disease (CAD) is an independent predictor of poor long-term outcomes [68]. Sinning et al. concluded that changes in sFlt-1 and PlGF levels did not predict the outcome of CAD when assessing cardiovascular mortality in patients with CAD using angiogenesis biomarkers [69]. In patients with acute coronary syndromes, the sFlt1/PlGF ratio did not improve risk stratification [70,71].

At least 10% of pregnant women have cardiovascular pathology [72]. Pregnancy alters the cardiovascular system, which must accommodate the increased metabolic demands of the mother and fetus [66,73]. The maternal cardiac output and heart rate increase, whereas the blood pressure and vascular resistance decrease. Because adaptive capabilities are reduced in pregnant women with certain heart diseases, pathological changes can occur at any stage of pregnancy, complicating its course and leading to adverse outcomes.

Despite the fact that PlGF is expressed at low levels in the heart and that the endothelium produces one of the splicing variants, sFlt-1 e15a, little is known about their role in a pregnant woman with cardiovascular pathology and how their level changes during pregnancy [54,74]. Angiogenic markers have been studied in pregnant women in the context of pre-eclampsia. It is known that pregnant women who develop pre-eclampsia have a higher risk (up to 4 times) of clinical CVD in the short and long term [75]. Angiogenic imbalance in pre-eclampsia is associated with a negative impact on the future risk of CVD in the long term after pregnancy. Lower PlGF levels during pregnancies complicated by pre-eclampsia were associated with poorer maternal hemodynamics (higher BP), subclinical myocardial dysfunction (worse global longitudinal strain), and an increased carotid intima-media thickness at 12 years postpartum [76]. Signs of arterial aging during pregnancy were positively correlated with higher sFlt-1 levels and a lower PlGF in women with pre-eclampsia. The sFlt-1 levels and the sFlt1/PlGF ratio were still greater in the pre-eclampsia group at one year postpartum and were connected to the degree of arterial aging [77]. 

Pregnancies complicated by hypertension, especially by severe pre-eclampsia, greatly increase the likelihood of the mother developing peripartum cardiomyopathy (PPCM) [78,79]. Mebazaa et al. discovered that women with PPCM had higher mean PlGF concentrations and lower sFlt1/PlGF ratios. The area under the curve for PlGF with a cut-off value of 50 ng/mL and/or the sFlt1/PlGF ratio with a cut-off value greater than 4 should be used to differentiate between PPCM and acute heart failure [80].

PlGF or sFlt-1 are thought to influence not only angiogenesis but also cardiac remodeling under pressure overload. A prospective cohort study by Ullmo et al. provided the first evidence describing the relationship between PIGF, sFlt-1, and their ratio and cardiovascular dysfunction/remodeling in women at risk of developing pre-eclampsia during and after pregnancy [81].

The link between cardiovascular health and pregnancy-associated PlGF and sFlt-1 abnormalities provides a theoretical basis for how these two proteins affect CVD in pregnant women, but the evidence is conflicting, and more research is needed (Figure 3).

## 9. sFlt1/PlGF Ratio and Rheumatoid Arthritis

Rheumatoid arthritis (RA) is a systemic autoimmune disease characterized by progressive joint destruction and damage to internal organs [82]. The prevalence of RA ranges from 0.4% to 1.3% [83,84]. RA is most common in women of childbearing age, and evidence suggests that the prevalence of RA in women is increasing [83,84,85]. Pregnant women with RA are at a significantly increased risk of adverse outcomes related to medication and disease activity [86]. 

PlGF and sFlt-1 levels are elevated in both the serum and synovial fluid in RA, and their levels in synovial fluid correlate with disease activity [84]. According to a study by Neuman et al., the sFlt1/PlGF ratio is not influenced by disease activity in pregnant women with RA, and a cut-off ≤38 can be used to rule out pre-eclampsia [87].

## 10. sFlt1/PlGF Ratio and Chronic Kidney Disease

Chronic kidney disease (CKD) is defined as kidney damage that lasts at least three months and manifests as structural or functional impairment [88]. CKD affects approximately 3–5% of pregnancies and is linked to poor pregnancy outcomes, such as pre-eclampsia [89,90]. Women with CKD are ten times more likely than women without CKD to develop pre-eclampsia, with pre-eclampsia affecting up to 40% of pregnancies in women with CKD. Because CKD, similar to pre-eclampsia, is associated with hypertension, proteinuria, and progressive renal failure, differential diagnosis during pregnancy may be difficult or impossible [91,92].

Several studies have been conducted to test the ability of the maternal sFlt1/PlGF ratio to distinguish CKD from pre-eclampsia [93,94,95,96]. The differential diagnosis of CKD during pregnancy and pre-eclampsia can be confirmed by analyzing sFlt-1, PlGF, and their ratio [94,95,96]. Pregnant women who have CKD with suspected superimposed pre-eclampsia, along with severe angiogenic imbalance (sFlt1/PlGF ≥ 85), had a higher incidence of confirmed superimposed pre-eclampsia compared to patients without or with mild angiogenic imbalance (sFlt1/PlGF ≤ 38 and sFlt1/PlGF > 38 to <85). The rate of progression to superimposed pre-eclampsia increased progressively as the degree of the angiogenic imbalance increased [93].

Despite the fact that the use of the sFlt1/PlGF ratio in clinical practice is still controversial today, it is considered one of the most promising biomarkers of pre-eclampsia. In the future, the use of the sFlt1/PlGF ratio would allow for the differential diagnosis of CKD with pre-eclampsia and the choice of the optimal course of treatment.

## 11. sFlt1/PlGF Ratio and Acute Fatty Liver of Pregnancy

AFLP is a rare disease. It is a potentially fatal condition characterized by hepatic failure during the third trimester of pregnancy, which is associated with multiorgan involvement and a variety of clinical and laboratory abnormalities [97]. The Swansea criteria have been suggested as an initial pilot diagnostic tool. AFLP is characterized by generalized malaise, nausea, vomiting, and abdominal pain, making early diagnosis difficult. There is a lower incidence of hypertension and proteinuria compared to pre-eclampsia, and patients with AFLP may develop encephalopathy with rapid progression to acute liver failure [98]. Pre-eclampsia-associated HELLP may be associated with liver dysfunction as well as AFLP. An sFlt-1 value above 31,100 pg/mL may be an additional biochemical feature, increasing discrimination between AFLP and HELLP syndrome, according to the proposed Swansea criteria to diagnose the condition. A linear correlation was discovered between the total number of Swansea criteria and sFlt-1 serum levels [99].

It is reasonable to expect that when pre-eclampsia is combined with AFLP, the level of sFlt-1 will rise dramatically, resulting in a high sFlt1/PlGF ratio and serious consequences for the mother and fetus.

## 12. sFlt1/PlGF Ratio and Obesity in Pregnant Women

Obesity is a chronic disease marked by an abnormal increase in body weight caused by adipose tissue. Obesity was prevalent in 15% of women [100]. Obesity has been shown to affect the levels of angiogenic markers in pregnant women, increasing the risk of pre-eclampsia development by 3–5 times [101,102,103,104]. Obese women with superimposed pre-eclampsia have higher sFlt1/PlGF levels than obese women without pre-eclampsia [105]. Due to a decrease in PlGF levels between the first and second trimesters of pregnancy, maternal weight was inversely associated with the sFlt1/PlGF ratio in the first trimester and positively associated with the sFlt1/PlGF ratio in the second trimester. The sFlt1/PlGF ratio in the second trimester occurred 3.13 times more frequently in pregnant women with grade II or III obesity than in pregnant women with normal weight [106]. sFlt1/PlGF is a useful predictor of adverse outcomes in normal, overweight, and obese women. However, its predictive value was limited due to the low incidence of adverse outcomes, including pre-eclampsia, in a relatively small cohort of obese women studied [107]. 

In obese women, the sFlt1/PlGF ratio can be used to predict negative outcomes. However, it is unclear which sFlt1/PlGF ratios should be used to predict and diagnose pre-eclampsia in obese women [108].

## 13. sFlt1/PlGF Ratio and Diabetes

Insulin resistance gradually rises at the beginning of the second trimester of a healthy pregnancy. The synthesis of progesterone, estrogens, and placental lactogen, which has a potent anti-insular impact, rises simultaneously with an increase in the hormonal activity of the placenta. Under the influence of estrogens, glucose is more actively utilized due to its passive transfer from mother to fetus. In the mother’s body, carbohydrate metabolism is compensated for by hypertrophy and hyperplasia of pancreatic β-cells, which increases insulin secretion and practically maintains blood glucose levels [109]. Variable levels of hyperglycemia occur in diabetes because the pancreatic beta cells are unable to respond appropriately to the increasing need for insulin. The risk of pre-eclampsia is greatly increased by diabetes since the placenta is vulnerable to hyperglycemia, which worsens vasculogenesis, angiogenesis, and microvascular remodeling (Figure 4) [110,111,112,113].

Pre-eclamptic women with pregestational type 1 or type 2 diabetes mellitus (T1DM or T2DM) had higher levels of sFlt-1, lower levels of PlGF, and higher sFlt1/PlGF ratios than pre-eclamptic women without pre-eclampsia, according to earlier research [114,115,116]. Women with T1DM or T2DM, who did not have pre-eclampsia or prenatal hypertension, had median sFlt1/PlGF levels at 35 to 40 weeks that were three times greater than those of controls who did not have diabetes. Even more significant alterations in the sFlt1/PlGF ratio occurred in diabetic women who later had pre-eclampsia [115]. Women with T1DM and pre-eclampsia exhibited greater sFlt-1 levels, lower PlGF levels, and a higher sFlt1/PlGF ratio, according to a study by Yu and colleagues [117]. Holmes et al. discovered that the ratio of sFlt1/PlGF considerably improved its predictive value in predicting the probability of developing pre-eclampsia in women with T1DM when combined with identified clinical risk variables and established clinical risk factors [118].

An investigation of the connection between gestational diabetes mellitus (GDM), pre-eclampsia, and placental biomarkers has, so far, only been done by Nuzzo et al. It was discovered that the ratio of sFlt1/PlGF in the blood serum of pregnant women with pre-eclampsia, and pre-eclampsia in the background of GDM (GDM-PE), was significantly greater when compared to healthy pregnant women and a group of pregnant women with GDM. Researchers have hypothesized that excessive sFlt-1 production in pregnant GDM-PE patients increases the risk of pre-eclampsia, but increased PlGF results in less severe endothelial dysfunction and a lower sFlt1/PlGF ratio [119].

Since changes in sFlt1 and PlGF levels appear before the development of the clinical signs of pre-eclampsia, the sFlt1/PlGF ratio can serve as a predictive marker for pre-eclampsia in pregnant women with GDM. However, further investigations are required to clarify its potential role in the early identification of GDM patients at risk of pre-eclampsia.

## 14. sFlt1/PlGF Ratio and Serum and Urine Biomarkers of Pregnant Women

Pregnant women may have biochemical changes in their blood serum before their sFlt1/PlGF ratio rises and pre-eclampsia develops [120,121].

A number of scientific studies have shown that pre-eclampsia increases not only the level of sflt1 but also the soluble endoglin (sEng) [15,36,122,123,124,125]. An increased level of sEng is usually accompanied by, and correlated with, an increased sFlt1/PlGF ratio [126,127]. Iannaccone et al. noted that the highest indicators of sFlt-1/PlGF ratio and sEng were observed in HELLP syndrome and EOPE [127].

Several earlier experimental studies looked at the relationship between angiogenic agents and other maternal serum parameters [123,128,129]. According to a study by Sarween et al., while the levels of free immunoglobulin light chains (sFLC), a highly sensitive C-reactive protein (HS-CRP), beta-2 microglobulin (B2-M), and sFlt1/PlGF were elevated in pre-eclamptic women, serum IgG (subclasses 1 and 3), albumin, and C4 complement protein were significantly lower in pre-eclamptic women than in the control group [128]. At 24 weeks of gestation, women with pre-eclampsia showed higher levels of sFlt1/PlGF ratio and blood uric acid [129].

Abascal-Saiz et al. identified a rise in blood levels of free fatty acids and C-peptide in pre-eclamptic moms and their link with the sFlt1/PlGF ratio on the basis that changes in lipid and carbohydrate metabolism are seen in women with pre-eclampsia [123]. In urine, the relationship between angiogenic agents and a few metabolic indicators has been studied. Urinary phthalate diesters have been linked to lower plasma levels of PlGF and a higher sFlt-1 to PlGF ratio in pregnant women. In the early stages of pregnancy, Philips et al. discovered subclinical relationships between urine phthalate metabolites and the sFlt1/PlGF ratio. The findings of this study provided significant evidence for the link between phthalate metabolites and the risk of pre-eclampsia [120]. 

Thus, women with pre-eclampsia showed significantly lower blood IgG, albumin, and C4 levels than controls, whereas total sFLC, HS-CRP, B2-M, and sFlt1/PlGF concentrations were higher [128]. Maternal plasma AT1-AA also increases in 26–60 weeks of gestation, complicated by pre-eclampsia [130]. Serum uric acid, mean total peripheral resistance (TPR), and phthalate are also correlated with both sFlt-1 and the sFlt1/PlGF ratio [120,129]. Early pregnancy measurements of these indicators may be useful for anticipating increased sFlt1/PlGF ratios, but further study is required.

## 15. sFlt1/PlGF Ratio and COVID-19

Given that pre-eclampsia and COVID-19 (SARS-CoV-2) have several medical characteristics, it is possible that the higher prevalence of pre-eclampsia among moms who had COVID-19 infections was the result of a misdiagnosis. In order to prevent unneeded procedures and induced preterm labor, pregnant women with COVID-19 may develop a pre-eclampsia-like syndrome that can be adequately identified by angiogenic markers. However, it must be highlighted that viral conditions can also affect the sFlt1/PlGF ratio and that uncontrolled amounts of those mediators are associated with placental insufficiency [131]. Since significant pathologic changes in the sFlt1/PlGF ratio cannot be detected during the symptomatic phase, Giardini et al. (2022) indicated that a COVID-19 infection may result in placental dysfunction if untreated. At the same time, sFlt-1 rose above 3000 pg/mL, however, in 11% of cases, the sFlt1/PlGF ratio was above 85 (110), and the patients developed pre-eclampsia, including placental dysfunction and FGR [132] (Figure 5). Additional research revealed that the sFlt1/PlGF ratio was unrelated to the severity of COVID-19 [132,133]. However, pregnant women with severe COVID-19 had greater levels of sFlt-1, which can be used to predict critical adverse pregnancy outcomes like severe pneumonia, intensive care unit hospitalization, viral sepsis, and maternal mortality [134].

In this regard, the presence of COVID-19 during pregnancy is a new additional factor for the development of pre-eclampsia.

## 16. sFlt1/PlGF Ratio and HIV Infection

Similar to COVID-19, HIV infection also contributes to angiogenesis issues during pregnancy, especially in the context of pre-eclampsia. COVID-19 infections occur concurrently with an HIV-associated pregnancy because both have an impact on the inflammatory response and endothelial function [135]. Because HIV is able to block endogenous antioxidant enzymatic systems, HIV-infected patients exhibit elevated systemic oxidative stress, similar to pre-eclampsia [136,137]. Recent research points to HIV-1 proteins as the primary cause of endothelial dysfunction [138]. Regardless of HIV status, pre-eclampsia was associated with higher levels of circulating sFlt-1 compared to normotensive pregnancies, despite a downward trend in sFlt-1 levels between HIV-infected and HIV-uninfected groups [139]. Pregnant people with HIV experience pre-eclampsia at a lower rate than pregnant people without HIV [140,141]; nevertheless, the risk of developing pre-eclampsia is disputed [141]. In HIV-associated pre-eclampsia, Govender et al. found no statistically significant difference between sFlt-1 and PlGF levels [139]. However, according to new data, in all cases of pregnancy-complicated HIV infection, sFlt-1 was elevated rather than PlGF being downregulated, according to research by Padayachee et al. [142]. 

## 17. sFlt-1, PlGF and Thyroid Hormones

The adaptive mechanisms of thyroid function provide the necessary hormonal conditions for a normal pregnancy. An adequate supply of thyroid hormones is essential for a smooth pregnancy and optimal fetal growth and development. Thyroid dysfunction in the mother during pregnancy is associated with an increased risk of adverse outcomes. Hypothyroidism during pregnancy was associated with an increased risk of pre-eclampsia [143].

PlGF and sFlt-1, which are produced by the placenta during pregnancy, might have an effect on the thyroid [144,145]. Levine et al. discovered that an increase in maternal sFlt-1 was associated with an increase in maternal thyroid-stimulating hormone (TSH) levels [144]. High PIGF levels in early pregnancy were linked to lower TSH and free thyroxine (FT4) levels, as well as an increased risk of isolated hypothyroxinemia [145]. Increased sFlt-1 levels have been linked to lower FT4 and thyroxine (T4) levels, as well as an increased risk of isolated hypothyroxinemia and subclinical hypothyroidism. This led them to the conclusion that high levels of PlGF and sFlt-1 in pregnant women at 18 weeks of gestation may be associated with a risk of maternal thyroid dysfunction.

Since there is little data on the relationship between the sFlt1/PlGF ratio and thyroid hormones, it is not possible to draw any conclusions regarding the relationship between the two indicators.

## 18. sFlt1/PlGF Ratio and In Vitro Fertilization (IVF)

Over 5 million children have been born as a result of assisted reproductive technologies (ART) procedures, including IVF since ART has been around for more than three decades. ART increases the incidence of obstetric problems compared to spontaneously conceived pregnancies, despite the fact that the majority of these pregnancies have a positive prognosis [146]. IVF procedures frequently lead to pregnancies without the corpus luteum (CL), which secretes vasoactive hormones [147]. A transient reproductive gland called the CL develops after ovulation and provides the progesterone required to start and sustain pregnancy [148]. Rapid growth, differentiation, and regulated regression are traits of the CL lifecycle. Intensive angiogenesis, stability, and angioregression are correlated with these phases. In early pregnancy, a lack of CL reduces maternal circulation and raises the risk of pre-eclampsia [149].

Since the imbalance of angiogenic factors plays a crucial role in the development of pre-eclampsia, Conrad et al., 2019 studied the sFlt1/PlGF ratio in patients after IVF. There was a dramatic increase in the plasma sFlt1/PlGF ratio between gestational weeks 5–6 and 5–7 of the first trimester. Free PLGF, on the other hand, gradually increased throughout this gestation. The sFlt1/PlGF ratio peaked during pregnancy at a value of about 100 in the first trimester [149]. Compared to spontaneously conceived fetuses, IVF-conceived pregnancies had higher levels of sFlt-1 and lower levels of PlGF at 18 and 35 weeks of gestation [150].

It can be assumed that in the absence of a CL, women who conceived by IVF may have pathophysiological changes in the placenta, however, the increased risk of developing a higher sFlt1/PlGF ratio remains unknown.

IVF treatment includes ovarian stimulation. The ovarian sensitivity index (OSI) is computed by dividing the total dose of exogenous follicle-stimulating hormone by the amount of retrieved oocytes [151]. When compared to the number of retrieved oocytes, OSI has been shown to be a better measure of ovarian responsiveness to gonadotropin. It can also be used as an index in studies with different ovarian stimulation doses during IVF [152]. Positive correlations between the PlGF/sFlt-1 ratio and the quantity of oocytes, embryos, and the OSI were discovered by Nejabati et al. The PlGF/sFlt-1 ratio could be used as a marker to identify high-responder women and could potentially allow for the identification of patients at risk for ovarian hyperstimulation syndrome. Follicular fluid soluble receptors for advanced glycation end product levels may also be a good predictor of the outcome of assisted reproduction technology [153].

## 19. sFlt1/PlGF Ratio and Cigarette Smoking

Smoking has been connected to a variety of detrimental impacts on human health, particularly during pregnancy [154,155]. Contrary to popular belief, there is a negative correlation between pregnancy-related cigarette smoking and the likelihood of developing pre-eclampsia [156]. Smokers’ levels of PlGF were considerably greater than those of non-smokers. According to this argument, smoking may have a protective effect against pre-eclampsia by raising levels of PlGF and the PlGF/sFlt-1 ratio [157]. Smoking has been linked to reduced maternal sFlt-1 levels during pregnancy complicated by pre-eclampsia, and smokers with normal pregnancies had considerably lower sFlt/PlGF ratios, but not when compared to non-smokers and smokers with pre-eclampsia [158].

Thus, cigarette smoking is not a risk factor for the development of pre-eclampsia but may reduce the risk of its development due to the modulation of angiogenesis proteins.

## 20. sFlt1/PlGF Ratio and Breast Cancer in Pregnant Women

Pregnancy-associated breast cancer (BC) refers to the detection of BC at the same time as the pregnancy or the detection of BC during lactation or no later than one year after the completion of the pregnancy [159]. BC occurs in about 1 in every 3000 pregnancies [160]. It is more common in women aged 32 to 38, with up to 7.3% of them pregnant or breastfeeding [161].

Angiogenesis activation is important in the growth and progression of BC [161,162] and may promote tumor growth, development, and metastasis, whereas antiangiogenic factors inhibit angiogenesis [163]. An imbalance of angiogenic and antiangiogenic factors occurs in pre-eclampsia: a high level of sFlt-1 is observed, which neutralizes the proangiogenic effects of PlGF, resulting in a decrease in tumor growth and development [164]. Pre-eclampsia reduces the risk of BC in the mother by 10–20% regardless of the timing and number of pregnancies and by four times in female offspring [165,166,167]. Sun et al. found no link between pre-eclampsia and the maternal risk of developing BC; however, pre-eclampsia has been linked to a significantly lower incidence of BC in women who have male offspring [168]. Numerous studies have failed to provide conclusive evidence that maternal hormones, sFlt 1, PlGF, or the future risk of maternal BC are linked [164,169]. Thus, the protective effect of pre-eclampsia on cancer incidence is not universal.

Depending on the stage of the disease and the age of the unborn child, chemotherapy is used as the standard treatment for BC diagnosed during pregnancy [170]. By analyzing levels of circulating angiogenic factors in pregnant women with BC treated with doxorubicin- and taxane-based chemotherapy regimens, it was evident that at the end of the third trimester, women with BC had significantly higher levels of sFlt-1 and ratios of sFlt-1/PlGF compared to the control group. Plasma levels of sFlt-1 were significantly correlated with the number of chemotherapy cycles [171]. The use of the sFlt1/PlGF ratio in clinical practice may be useful in the management of pregnant women with BC who are receiving chemotherapy, as it can predict the placental complications that these patients typically develop.

## 21. Conclusions

Maternal illnesses may increase the risk of developing pre-eclampsia. Markers of angiogenesis, including the sFlt1/PlGF ratio, are well-studied in the field of pre-eclampsia. The analysis obtained in the literature showed that the disease incidence significantly increased the level of sFlt-1 and significantly increased the level of sFlt1/PlGF. 

These factors are maternal pregestational comorbidities, which are associated with increasing sFlt-1 or decreasing PlGF levels. New risk factors for an increased sFlt1/PlGF ratio include lower blood IgG, albumin, C4 levels, and larger sFLC, HS-CRP, B2-M, TPR, phthalate, COVID-19, AFLP, peripartum cardiomyopathy, and aberrant myocardial performance and OSI. Other factors, such as HIV, acute coronary syndromes, maternal cardiovascular function at 19–23 weeks gestation, thyroid hormones, diabetes, and cancer, remain contradictory. These data require further confirmation using case-control studies, which will take into account not only the sFlt-1/PlGF ratio, the condition of the mother of the fetus during and after delivery, but also the presence of chronic diseases in the mother (especially the lungs, muscles, and female reproductive organs and skin, where the *FLT1* gene is expressed) (Figure 6).

## Figures and Tables

**Figure 1 ijms-24-06744-f001:**
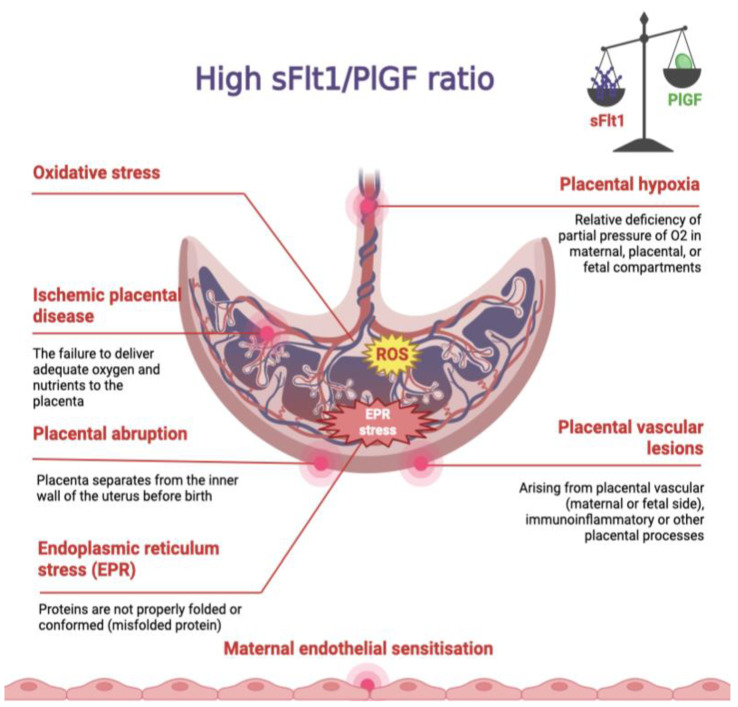
The figure summarizes the various pathological changes typical of pre-eclampsia, which are associated with an increasing sFlt1/PlGF ratio.

**Figure 2 ijms-24-06744-f002:**
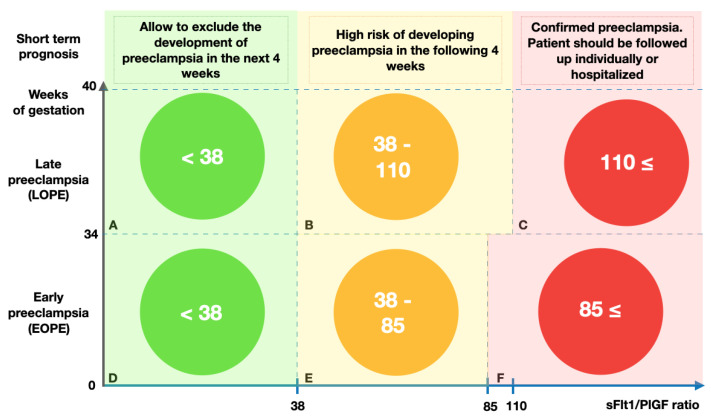
Using the sFlt1/PlGF ratio for short-term prediction and confirmation of pre-eclampsia. (**A**). Week of gestation > 34 and the sFlt1/PlGF ratio < 38 allow excluding the development of LOPE in the next 4 weeks. (**B**). Week of gestation > 34 and the sFlt1/PlGF ratio 38–110 indicate a high risk of developing LOPE in the following 4 weeks. (**C**). Week of gestation > 34 and the sFlt1/PlGF ratio ≥ 110 allow confirming preeclampsia. (**D**). Week of gestation < 34 and the sFlt1/PlGF ratio < 38 allow excluding the development of EOPE in the next 4 weeks. (**E**). Week of gestation < 34 and the sFlt1/PlGF ratio 38–85 indicate a high risk of developing EOPE in the following 4 weeks. (**F**). Week of gestation < 34 and the sFlt1/PlGF ratio ≥ 85 allow confirming preeclampsia.

**Figure 3 ijms-24-06744-f003:**
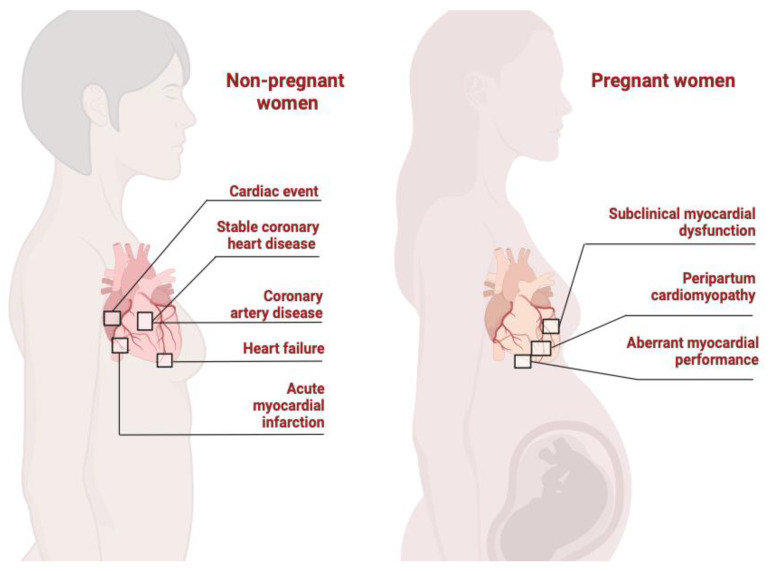
Illustrates interaction between sFlt1/PlGF ratio and cardiovascular disease in unpregnant and pregnant women.

**Figure 4 ijms-24-06744-f004:**
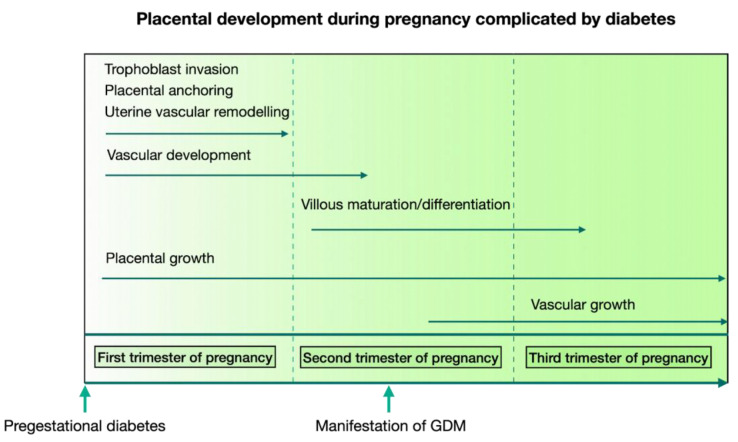
Illustration of placental development during pregnancy with designation of the timing of manifestation of pregestational diabetes (DM1, T2DM) and GDM.

**Figure 5 ijms-24-06744-f005:**
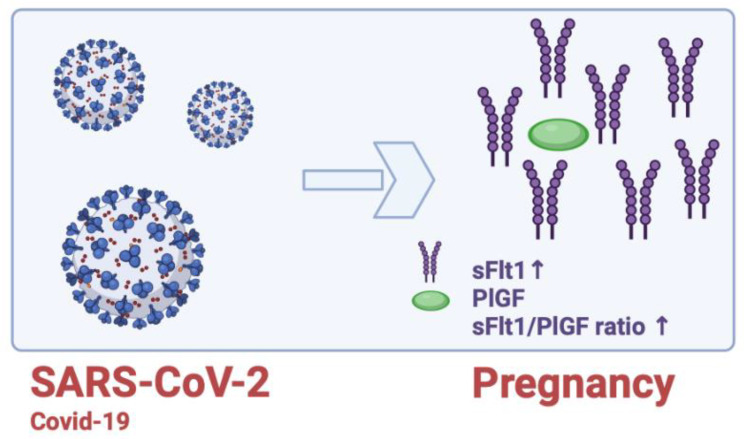
Impact of COVID-19 on the sFlt1/PlGF Ratio.

**Figure 6 ijms-24-06744-f006:**
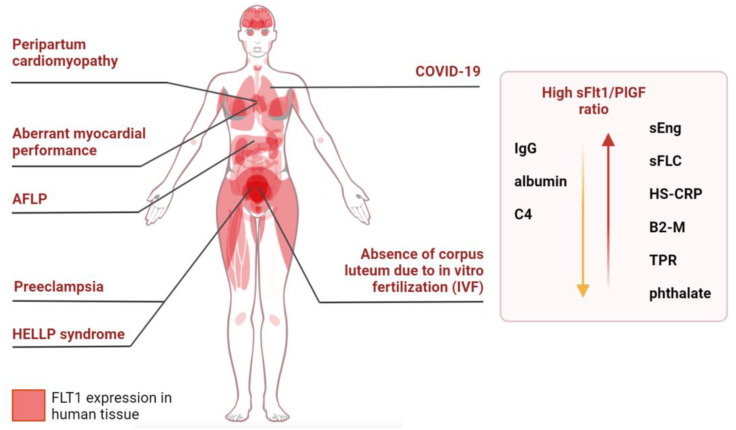
Anatomogram of *FLT1* expression in human tissues with maternal pregestational comorbidities that may be associated with increased sFlt1/PlGF ratios during pregnancy causing pre-eclampsia.

## Data Availability

Not applicable.

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
