# Peer review of "Literature Review: The sFlt1/PlGF Ratio and Pregestational Maternal Comorbidities: New Risk Factors to Predict Pre-Eclampsia"

_ijms, 2023, doi:10.3390/ijms24076744_

Round 1
Reviewer 1 Report
This is an interesting review, which summarizes the use of ratio sFlt-1/PlGF in obstetrics. The ratio is a very potent biomarker not only in the field of PE, but also other placental syndromes like FGR or PA. Circulating levels of angiogenic factors in pregestational mother comorbidities is a totally worth further analysis topic.
However, some issues must be changes before it is published.
Chapter 1. First of all, the definition of PE must be updated – the definition formed in 2002 is now more than 20 years old and since 2018 the broad ISSHP definition is respected, also by ACOG which renewed its criteria in 2020.
Chapter 2. It is titled sFlt-1 &PlGF but it contains only the description of PlGF. PlGF is defined by the Authors as cytokine but is wrong; is an pro angiogenic protein, that may be produced by different tissues under certain condition, but it does not meet the definition od cytokine. Cytokines are produced mostly by immune cells. Cytokines include chemokines, interferons, interleukins, lymphokines, and tumour necrosis factors, but generally not hormones or growth factors (despite some overlap in the terminology).This introduction must be changed.
Chapter 3 I am not sure, whether we can define changes in Flt-1/PlGF ratio as a reason of ectopic pregnancy. It should be clarified what you mean. Also, the data about prediction of placental abruption (PA) is still being discussed and there are many analysis, and metaanalysis that confirm the sFlt-1/PlGF ratio connection with the prediction of adverse outcomes including PA. The association between maternal angiogenic factors and the risk of placental abruption has been examined in several studies and systematic reviews, that should be taken into consideration by the authors.
Raw 86 –the name of the author is ZAISLER not ZAISIER
Raw 104 – miscarriage is not the right word
Raw 108 – construction of the sentence
Raw 133 – not true, elevated ratio is not specific to PE, as the authors mentioned before, levels of the ratio are as high an in PE in FGR cases (studies by Herraiz, Alahakoon, Dymara-Konopka)
Raw 140-141 construction of sentence is not clear
Raw 142-143 Maternal risk factors of PE (underlying medical conditions) are already defined, now we ware looking for biomarkers that help us to identify and monitor evolution of the syndrome
Raw 149 construction of sentence is not clear
Raw 151-152 HIGHER RATIO COMPARING TO WHAT??? The performance of sFlt and PlGF has been demonstrated in differential diagnosis between PE and other diseases that can mimic PE, particularly before 34 weeks. These include gestational hypertension and chronic hypertension (sometimes unnoticed earlier than 20 weeks of PE due to typical fall in blood pressure in early pregnancy), and this is the role of angiogenic biomarkers in here.
Mayor english revisions are needed.
Author Response
Thank you for your valuable and helpful corrections! We have updated the definition of preeclampsia according to the latest ACOG 2020 recommendations:
"Preeclampsia is a disorder of pregnancy associated with new-onset hypertension, which occurs most often after 20 weeks of gestation and frequently near term. Although often accompanied by new-onset proteinuria, hypertension and other signs or symptoms of preeclampsia may present in some women in the absence of proteinuria"
Chapter 2
Thank you for noticing the absence of a text fragment! Apparently we didn't transfer it from the draft of the manuscript. Now we have added the text about sFlt 1 to the text of the article.
"Antiangiogenic sFlt-1 encoded by the FLT1 gene located on chromosome 13. There are several splice variations in total sFlt-1, moreover, the e15a is expressed by the placenta, inhibits VEGF and significantly elevated in patients with preeclampsia. The placenta appears to be the main source of elevated sFlt-1 in preeclampsia as within 48 h following birth, circulating levels dramatically reduce [9]."
We also corrected the definition for PlGF.
Chapter 3
We explained that sFlt1/PlGF ratio can lead to ectopic pregnancy and missed abortion, but do not necessarily lead to it. Regarding the relationship of high sFlt1/PlGF ratio with placental abruption. Preeclampsia is directly associated with placental abruption and as part of the review, we do not consider separately the typical pathophysiological changes characteristic of preeclampsia, since the appearance of any of them increases the risk of developing preeclampsia and this has been known for a long time, and most of them are associated with high sFlt1/PlGF ratio.
Raw 86 - corrected
Raw 104 – miscarriage changed on late abortion
Raw 108 – The text of the article has been sent for verification and revision to a native speaker
Raw 133 – The incorrect part of the sentence was removed
Raw 140-141 The text of the article has been sent for verification and revision to a native speaker
Raw 142-143 The incorrect part of the sentence was removed
Raw 149 The text of the article has been sent for verification and revision to a native speaker
Raw 151-152 The results of studies by Perni et al. show higher sFlt1/PlGF ratios in pregnant women with superimposed EOPE compared to EOPE without CH
Reviewer 2 Report
Your article is a good overview of the conditions where we can expect to be affected by an altered sFlt1/PlGF ratio. In 20 points you have illustrated the different medical conditions and specific mechanisms in relation to pre-eclampsia.
In subsection number 14 you listed some biomarkers in serum or urine that might be altered before the ratio is changed.
In this chapter, I am missing information on Inhibin A and on Endoglin. We have many publications on this topic and I suggest that you include them in your text.
There is one more typo: in line 322, the word "and" is used twice.
Author Response
Good afternoon! Thank you for your comments. We checked the information and found that inhibin A does decrease in preeclampsia, but we did not find any data on its correlation with the sflt1/PlGF ratio. There are only data on changes in both the sflt1/PlGF ratio and inhibin A in individuals with preeclampsia, but there are not enough data [PMID: 18771979] to draw reliable conclusions. Ratio We included in the review only markers that correlate not only with preeclampsia, but also sflt1/PlGF. To study inhibin A in the context of various ratios, the sflt1/PlGF ratio will be very interesting in the future, but not within the framework of the current review. We have added information about sEng in the text.
"A number of scientific studies have shown that preeclampsia increases not only the level of sflt1, but also soluble endoglin (sEng) [123-128]. The increased sEng level was usually accompanied by an increased sFlt1/PlGF ratio and correlated with each other [129-130]. Iannaccone and et.al . It is noted that the highest sFlt-1/PlGF and sEng ratios are observed in HELLP and EOPE syndrome [130]."
We are also correcting a typo in line 322.